# Sociodemographic determinants of multidrug-resistant tuberculosis in Lesotho: A case-control study

Jerry Yakubu Yahaya ®*

Technical Services Division, Ghana AIDS Commission, Accra, Ghana

* jerrywayo123@gmail.com

## Abstract

The emergence of multidrug-resistant tuberculosis (MDR-TB) significantly undermines global efforts toward tuberculosis (TB) control, particularly in high-burden settings like Lesotho. Understanding the sociodemographic factors contributing to MDR-TB is crucial yet remains under-explored in this context. This study aimed to identify key sociodemographic determinants associated with MDR-TB among adult TB patients in Lesotho. Using a retrospective case-control design, I analyzed data from 306 participants, including confirmed MDR-TB cases and drug-susceptible TB controls, recruited from 12 TB clinics between March 2021 and February 2022. Sociodemographic characteristics (age, sex, education, employment, income, place of residence), HIV status, and caregiver presence were examined using chi-square tests and multivariable logistic regression analyses. The findings indicated that individuals older than 26 years had lower odds of MDR-TB compared to those aged 18–26 years (OR = 0.8, 95% CI 0.67–0.99, $p = 0.040$). Similarly, higher income levels (earning more than $1,026 annually) were associated with reduced odds of MDR-TB (OR = 0.5, 95% CI 0.22–0.94, $p = 0.034$). Conversely, the absence of caregiver support significantly increased the likelihood of MDR-TB by 80% (OR = 1.8, 95% CI 1.04–3.11, $p = 0.036$). These findings highlight the critical need for targeted interventions focusing on socioeconomic empowerment, caregiver support, and tailored public health education to effectively mitigate the MDR-TB burden in Lesotho.

## Introduction

### Context and background

Multidrug-resistant tuberculosis (MDR-TB) is a critical global health threat, characterized by resistance to at least isoniazid and rifampicin, the two most potent first-line anti-TB drugs [1]. MDR-TB complicates treatment, prolongs infectious periods, and significantly increases mortality rates [2,3]. Globally, MDR-TB accounts for approximately 3–4% of new TB cases and 18–21% of previously treated cases, posing a

**Data availability statement:** The de-identified dataset used in this study has been made publicly accessible on Figshare to support transparency and reproducibility. It can be accessed using the following link: https://figshare.com/s/40c110742dd6e66d5ff7 Digital Object Identifier (DOI): 10.6084/m9.figshare.29263616

**Funding:** The research presented in the manuscript titled "Sociodemographic Determinants of Multidrug-Resistant Tuberculosis in Lesotho: A Case-Control Study" was fully supported by personal funds. No external funding, grants, or financial support from any organization or institution was received for the design, data collection, analysis, or preparation of this study.

**Competing interests:** The authors have declared that no competing interests exist.

significant challenge to TB control efforts [4]. Lesotho, one of the highest TB burden countries, has an MDR-TB incidence rate of 41 per 100,000 population, making it a major public health concern [5]. The country's TB control efforts are further complicated by a high HIV co-infection rate, which exacerbates disease progression and treatment challenges [6]. The Ministry of Health's TB and Leprosy Programme has implemented national strategies to detect and treat 90% of new TB cases, achieve a 90% cure rate, and improve diagnostic services to enhance treatment outcomes [7]. Additionally, targeted community-based interventions and rigorous contact tracing have been employed to strengthen TB case detection and management, particularly in rural and underserved areas [8].

Despite these efforts, MDR-TB remains a pressing concern, with socioeconomic and demographic factors playing a crucial role in disease transmission and treatment adherence. Factors such as poverty, inadequate healthcare access, and high rates of HIV co-infection contribute to the persistence of MDR-TB in Lesotho. Understanding the sociodemographic determinants influencing MDR-TB is essential for tailoring effective public health interventions. This study aims to examine these factors to inform evidence-based strategies for MDR-TB prevention and control in Lesotho.

### Objective/research question

This study examined the complex interplay of the factors contributing to multidrug-resistant tuberculosis (MDR-TB) in Lesotho, a high-burden setting. Specifically, the objectives were twofold: first, to quantitatively determine the association between key sociodemographic variables, including age, sex, income, education, employment status, and place of residence, and the likelihood of MDR-TB diagnosis amongst tuberculosis patients; and second, to rigorously assess the independent and synergistic contributions of HIV co-infection to the development and potential transmission of MDR-TB within this population. The findings of this research is aimed to generate robust, evidence-based data to inform the development of targeted public health interventions, ultimately aimed at reducing the burden of MDR-TB in Lesotho and similar high-burden contexts.

### Rationale

The significance of this research lies in its potential to fill critical knowledge gaps in understanding the risk factors for MDR-TB in Lesotho. While there is extensive literature on TB and MDR-TB in other settings, few studies have examined the specific socioeconomic and demographic variables influencing MDR-TB in Lesotho. MDR-TB is one of the most pressing challenges in global TB control, and understanding the underlying factors specific to Lesotho is essential for improving disease management, treatment outcomes, and prevention strategies. Additionally, the findings from this research will assist policymakers and healthcare providers in Lesotho in developing targeted interventions to mitigate the identified risk factors. These insights can be adapted to similar settings in other developing countries, contributing to the global effort to end TB as outlined in the WHO's End TB Strategy.

## Materials and methods

### Ethics

Ethical approval for this study was obtained from the Walden University Institutional Review Board (IRB) and the Lesotho Health Services. Informed consent was obtained from all participants before data collection. Participants were informed of their rights to withdraw at any time without penalty, and their confidentiality was maintained using unique study identification numbers. All data were anonymized, and no identifiable information was included in the analysis or reporting of results. The data were stored in a secure, password-protected database accessible only to the principal investigator.

### Study design

This study employed a retrospective case-control design to investigate the association between sociodemographic factors and the risk of multidrug-resistant tuberculosis (MDR-TB) in Lesotho. The case group includes confirmed MDR-TB patients aged 18 years and above, diagnosed at any of the 12 TB clinics across Lesotho between March 2021 and February 2022. The control group consists of individuals diagnosed with pulmonary TB (non-MDR-TB) who share similar sociodemographic characteristics, including age, employment, income, sex, education, and place of residence [9,10]. A total of 306 participants were included in the final analysis, with 115 (37.6%) assigned to the case group (MDR-TB) and 191 (62.4%) to the control group (pulmonary TB without multidrug resistance). The distribution of participants was determined based on the epidemiological data, which indicated that pulmonary TB cases were approximately twice as prevalent as MDR-TB cases in the study population. This allocation ensured a representative comparison between individuals with drug-resistant and drug-susceptible TB, allowing for a robust assessment of sociodemographic and HIV-coinfection as risk factors associated with MDR-TB. The case-control design allowed the identification of potential risk factors for MDR-TB by comparing exposed and non-exposed individuals, particularly in populations with relatively low incidence rates of the disease [11]. Furthermore, it enabled an analysis of the interplay between HIV co-infection and MDR-TB, providing insights into their combined impact on disease burden, thus informing targeted public health interventions.

### Study population

The study population included individuals who received TB treatment at referral centers in Lesotho between January 2018 and January 2022. Participants were adults (18 years and older) with confirmed TB cases. The inclusion criteria encompassed adult outpatients with either pulmonary TB or MDR-TB, who had a working knowledge of Sesotho or English, and were listed on the TB register. Exclusion criteria included individuals under 18 years, those with unconfirmed TB, or patients who were not receiving treatment during the study period.

### Sampling strategy

The sampling strategy used in this study involved a proportionate stratified random sampling approach to ensure adequate representation from multiple TB clinics across Lesotho. Initially, the target population included all adult patients (18 years or older) actively receiving treatment for confirmed TB and/or MDR-TB listed on the TB registers of the selected clinics. Participants not currently on TB treatment, below 18 years of age, or those without confirmed TB were excluded. The eligible population was divided into distinct subpopulations (strata) based on their community locations. From each stratum, individuals were randomly selected in proportion to the size of the stratum relative to the entire population. Participants were randomly identified through the assignment of numbers using Microsoft Excel-generated random numbers, ensuring each TB patient had an equal chance of selection. The calculated sample size, powered at 80% using G*Power software, was initially determined to be 186 participants, with a contingency added for potential missing data and multiple comparisons, leading to a final sample size of 306 participants. This methodological approach provided a robust foundation for statistical inference and enhanced the representativeness and generalizability of the findings within this high-burden setting.

**Global Public Health** PLOS

## Study variables

The primary outcome variable in this study was tuberculosis (TB) status, specifically the presence or absence of multidrug-resistant tuberculosis (MDR-TB). The independent variables encompassed a range of sociodemographic and health-related factors that were hypothesized to influence MDR-TB risk.

**Dependent variable.**

- TB Status: Categorized as either MDR-TB (presence) or non-MDR-TB (absence).

**Independent variables.**

- Educational Level: Classified into five categories: never attended school, elementary, high school, college, and higher education.

- Employment Status: Defined as employed, not employed, or retired.

- Income Level: Categorized as low, lower-middle, upper-middle, or high.

- Place of Residence: Dichotomized as rural or urban.

- *Sex*: Defined as male or female.

- Age Categories: Grouped into five intervals: 18–29, 30–39, 40–49, 50–59, and ≥60 years.

- HIV Status: Classified as positive or negative. Lesotho has one of the highest TB-HIV co-infection rates, with approximately 73% of TB patients being HIV-positive underscoring the critical interplay between these infections [4].

**Covariates:** In addition to the primary independent variables, the study incorporated key covariates to account for potential social and behavioral influences on MDR-TB risk. These include:

- Multiple Sexual Partners: Categorized as Yes/No, included as a potential exposure to HIV, a known risk factor for TB progression and drug resistance due to immunosuppression [12].

- Caregiver Support: Defined as Yes/No, indicating whether the participant had a caregiver, such as a family member. The presence of a caregiver has been associated with improved TB treatment adherence and lower mortality rates [13]. Including these covariates in the multivariate analysis allowed for a more comprehensive assessment of the sociodemographic and behavioral determinants of MDR-TB in Lesotho, ensuring that study findings accounted for key contextual factors influencing disease outcomes.

## Data collection

Data collection was carried out using a structured survey questionnaire, adapted from the Lesotho Demographic and Health Surveys (LDHS). The instrument was pretested for reliability and validity, and data collectors were trained accordingly. Participants were recruited through phone calls or during clinic visits. Information was gathered on participants' sociodemographic characteristics, health status, and risk behaviors through phone surveys and face-to-face interviews. The data collected were then cleaned and entered into a secure database for analysis.

## Data analysis

Sample size estimation was powered using G*Power software, which indicated that a sample of 186 participants would provide sufficient power (80%) for detecting an odds ratio of 2.3, with a significance level of 0.05. To account for missing data, a contingency of 25% was added, yielding a final sample of 306 participants. Data analysis involved Chi-Square test and logistic regression to assess the relationship between sociodemographic factors and the risk of MDR-TB, providing

insights to guide targeted public health interventions. Odds ratios (OR) were calculated for the effect sizes of the predictor variables. Statistical analysis was conducted using STATA software.

## Results

### Main findings

The sociodemographic characteristics of the study participants illustrate a diverse sample, with a total of 312 participants, of whom six (2.2%) were excluded from the final analysis due to the ages being below 18 years. Although the sample was randomized, the final sample comprised 200 females (65.4%) and 106 males (34.6%), reflecting a female predominance. This gender distribution aligns with findings from Luba et al. [14], who reported that 71.6% of tuberculosis cases in the 2014 Lesotho Demographic and Health Survey (LDHS) were among females. Various socio-cultural factors, gender roles, and other barriers might place women at greater risk of TB in Lesotho, as most caregivers in this context were women. The age of participants ranged from 18 to 98 years, with a mean age of 47.8 years ($SD = 16.2$), and the largest age group was 30–39 years (26.8%), followed by those who were 65 years or older (26.5%). This age distribution differs slightly from the 2014 LDHS, which reported the highest TB prevalence among individuals aged 15–24 years (44%). A majority of the participants resided in rural areas (73.2%), consistent with the 2014 LDHS, which found that 67.3% of the population lived in rural regions [14]. Education levels varied, with most participants completing elementary school (55.6%), while only a small fraction (1.0%) attained higher education. Employment status was notably skewed, with 74.5% of participants unemployed. Income disparities were also evident, with 85.6% of respondents classified as low-income. In terms of health conditions, 42.2% of participants were HIV-positive, and 37.6% had multidrug-resistant tuberculosis (MDR-TB), with 62.4% diagnosed with only pulmonary TB. Healthcare access remained a concern, as 15.4% of participants lived more than 10 km from a health facility, potentially impacting timely medical intervention and treatment adherence. These baseline characteristics are presented in Table 1, below. The characteristics provide critical context for understanding subsequent analyses of MDR-TB risk factors among the study population.

### Statistical results

A logistic regression model was developed to identify factors associated with multidrug-resistant tuberculosis (MDR-TB) among TB patients in Lesotho. Variables entered into the model include age, sex, income level, education, place of residence, employment status, HIV status, presence of multiple sex partners, and caregiver status. Variables demonstrating a $p$-value < .25 in the bivariate analyses were included in the multivariate logistic regression analysis to adjust for potential confounding.

Model fit was assessed using the Hosmer–Lemeshow goodness-of-fit test, resulting in a non-significant test statistic ($\chi^2 = 6.24$, $p = .621$), indicating adequate model fit. Additionally, the model demonstrated a pseudo $R^2$ value of 0.04, meaning approximately 4% of the variation in MDR-TB status could be explained by the variables included in the model. The Akaike Information Criterion (AIC = 397.37) and Bayesian Information Criterion (BIC = 438.07) further supported the selected model.

The logistic regression analysis revealed significant associations between MDR-TB and age, income, and caregiver support. The results of the variables entered in the model are detailed in Table 2, below. In contrast, sociodemographic factors such as education, employment status, gender, place of residence did not show significant associations with MDR-TB, nor did HIV status as a health-related determinant or the number of sexual partners as a behavioral factor.

The adjusted odds ratios (aOR) from the model revealed that participants aged 65 years and older had a significantly lower likelihood of MDR-TB compared to the reference category of younger participants aged 18–29 years (aOR = 0.80, 95% CI [0.67, 0.99], $p = .040$). Participants with higher income (earning more than $1,026 annually) showed significantly lower odds of developing MDR-TB compared to lower-income individuals (reference category) (aOR = 0.50, 95% CI [0.22,

**Table 1. Sociodemographic characteristics of study participants.**

| Characteristic | n (%) |
|---|---|
| Age Group (years) | |
| 18–29 | 36 (11.8%) |
| 30–39 | 82 (26.8%) |
| 40–49 | 51 (16.7%) |
| 50–59 | 56 (18.3%) |
| 65+ | 81 (26.5%) |
| Gender | |
| Male | 106 (34.6%) |
| Female | 200 (65.4%) |
| Geographic Setting | |
| Urban | 82 (26.8%) |
| Rural | 224 (73.2%) |
| Education Level | |
| Never attended school | 19 (6.2%) |
| Elementary school | 170 (55.6%) |
| High school | 96 (31.4%) |
| College | 18 (5.9%) |
| Higher education | 3 (1.0%) |
| Employment Status | |
| Not employed | 228 (74.5%) |
| Employed | 51 (16.7%) |
| Retired | 26 (8.5%) |
| Not answered | 1 (0.3%) |
| Income Status (USD) | |
| Below $1,026 (low income) | 262 (85.6%) |
| $1,026–$3,995 (lower middle income) | 37 (12.1%) |
| $3,995–$12,375 (upper middle income) | 3 (1.0%) |
| ≥$12,376 (high income) | 4 (1.3%) |
| HIV Status | |
| Negative | 177 (57.8%) |
| Positive | 129 (42.2%) |
| Number of Sexual Partners | |
| No partner | 25 (8.2%) |
| One partner | 127 (41.5%) |
| More than one partner | 154 (50.3%) |
| Distance to Health Facility | |
| Less than 2km | 113 (36.9%) |
| Between 2km and 5km | 77 (25.2%) |
| Between 6km and 10km | 69 (22.6%) |
| More than 10km | 47 (15.4%) |
| Have Caregiver | |
| No | 223 (72.9%) |
| Yes | 83 (27.1%) |
| MDR-TB Status | |
| No | 191 (62.4%) |

**Table 2. Logistic regression analysis of MDR-TB status and sociodemographic factors.**

| Characteristic | Unadjusted OR | p-value | 95% Confidence Interval | Adjusted OR | p-value | 95% Confidence Interval |
|---|---|---|---|---|---|---|
| Age group | 0.9 | 0.174 | [0.75, 1.05] | 0.8 | 0.040 | [0.67, 0.99] |
| Gender | 1.3 | 0.342 | [0.78, 2.08] | 1.4 | 0.205 | [0.83, 2.42] |
| Place of Residence | 0.8 | 0.397 | [0.48, 1.34] | 0.7 | 0.238 | [0.40, 1.25] |
| Education | 0.9 | 0.647 | [0.68, 1.27] | 1.1 | 0.685 | [0.73, 1.61] |
| Employment Status | 0.9 | 0.737 | [0.65, 1.35] | 1.2 | 0.379 | [0.79, 1.87] |
| Income | 0.6 | 0.076 | [0.34, 1.06] | 0.5 | 0.034 | [0.22, 0.94] |
| HIV Status | 1.1 | 0.716 | [0.68, 1.74] | 1.1 | 0.720 | [0.65, 1.85] |
| Multiple Sex Partners | 0.9 | 0.723 | [0.65, 1.35] | 0.8 | 0.355 | [0.56, 1.23] |
| Caregiver support | 1.8 | 0.032 | [1.05, 2.95] | 1.8 | 0.036 | [1.04, 3.11] |

*Pseudo R²=0.04; AIC=397.37; Hosmer–Lemeshow goodness-of-fit test: χ²=6.24, p=0.6205.*

0.94], *p*=.034). Conversely, not having a caregiver support was significantly associated with an increased likelihood of MDR-TB compared to those who had caregiver support (reference category) (*aOR* = 1.80, 95% CI [1.04, 3.11], *p*=.036).

## Discussion

### Interpretation of results

The findings of this study provide important insights into the sociodemographic determinants of multidrug-resistant tuberculosis (MDR-TB) in Lesotho, particularly in relation to age and income, and caregiver support as a covariate. The results demonstrated that younger TB patients, particularly those between the ages of 18–29, are at a significantly higher risk of developing MDR-TB compared to older patients, potentially due to non-compliance with treatment regimens and active lifestyles. Specifically, older people have a 20% lower likelihood of having MDR-TB compared to younger participants (18–29 years). This finding aligns with prior studies conducted in Lesotho and other regions, which similarly identified younger populations as being at heightened risk for MDR-TB [15–17]. However, some studies have found the opposite, with older age being a risk factor, possibly due to weakened immune systems [18–21]. Additionally, participants who belonged to a higher socioeconomic class were less likely to develop MDR-TB infection compared to those with lower incomes. Participants who earned more than $1,026 annually were associated with a significantly lower risk of MDR-TB. In other words, participants with higher incomes had a 50% reduced likelihood of MDR-TB infection compared to those with low-incomes. reinforcing the established connection between socioeconomic status and tuberculosis risk [2,10,22–25]. Undeniably, patients already experiencing poverty are particularly vulnerable to falling into deeper impoverishment following a diagnosis of tuberculosis (TB) or multidrug-resistant tuberculosis (MDR-TB). This is supported by findings from a study conducted in Indonesia, which reported that 32% of TB patients and an additional 69% of MDR-TB patients who were previously employed lost their jobs post-diagnosis [26]. Beyond the immediate economic impact on affected individuals and their families, the substantial costs associated with TB care can pose significant barriers to accessing and adhering to treatment. Variable expenses, including travel, food, nutritional supplements, and lost income, contribute considerably to the total economic burden faced by patients and their families. Consequently, it can be agreed from the results that economic heterogeneity significantly contributes to disparities in MDR-TB incidence in Lesotho.

Furthermore, caregiver support was significantly associated with MDR-TB (p=0.031), indicating that caregiver support reduces the risk of MDR-TB infection and patients without caregiver support were more likely to have MDR-TB. Not having caregiver support was associated with an 80% increased likelihood of MDR-TB compared to participants who had a caregiver support, emphasizing the critical role that family and community support systems play in ensuring treatment adherence and patient recovery [27]. The study did not find a significant association between HIV status and MDR-TB,

which is consistent with several studies [16,28,29] but contrasts with others that observed a stronger link between HIV and MDR-TB [30–32].

### Strengths and limitations

A key strength of this study is its rigorous case-control design and the use of a randomized sample drawn from 12 diverse TB clinics across Lesotho, enhancing the generalizability of findings to the broader TB patient population within the country. Additionally, the sample size was large enough for the results and findings to be generalized on the general population of the study. Moreover, this study addresses a critical gap in existing literature by specifically examining sociodemographic determinants of multidrug-resistant tuberculosis (MDR-TB) within a resource-limited context. However, several limitations warrant consideration. Firstly, reliance on self-reported data introduces the possibility of recall bias and social desirability bias. Future studies should consider triangulating self-reported information with clinical records or biomarkers to validate responses and mitigate these biases. Secondly, the study did not control for some potentially important confounding variables, including alcohol and tobacco use, as well as treatment adherence. Subsequent research should incorporate these variables into the study design, using detailed questionnaires or structured clinical interviews to thoroughly assess these additional factors. Lastly, while findings from this study are highly relevant for Lesotho, the unique socioeconomic conditions of the country may limit generalizability to other regions. Conducting similar studies in other geographic contexts would further clarify how varying socioeconomic and cultural contexts influence MDR-TB risks globally.

### Implications

The results from this study have significant implications for public health policy and practice. Targeted interventions should prioritize younger TB patients by enhancing awareness and education regarding MDR-TB, alongside improving healthcare accessibility for economically disadvantaged populations. Furthermore, caregiver roles must be strengthened through focused public health initiatives emphasizing training on infection control practices and effective patient management strategies. From a policy perspective, integrating social protection programs, such as conditional cash transfers, within TB care frameworks may mitigate economic barriers faced by low-income individuals, enhancing treatment access, and subsequently reducing MDR-TB transmission. Additionally, the absence of a substantial link between HIV status and MDR-TB risk in this study indicates current HIV management practices may not require significant modifications; however, ongoing research remains necessary to further clarify this association in diverse contexts. Additionally, the absence of a substantial link between HIV status and MDR-TB risk in this study indicates current HIV management practices may not require significant modifications; however, ongoing research remains necessary to further clarify this association in diverse contexts.

### Conclusion

This study highlights the significant role of age, income, and caregiver support in moderating the risk of MDR-TB among TB patients in Lesotho. Younger age groups, individuals with lower economic security, and those without caregiver support emerged as particularly vulnerable populations. Enhancing socioeconomic conditions and strengthening caregiver support through targeted policies and public health initiatives will be vital in mitigating MDR-TB transmission and improving overall TB management outcomes. Further research exploring additional behavioral and sociocultural factors is essential for developing comprehensive strategies for MDR-TB prevention and control in Lesotho and similar high-burden settings.

### Supporting information

**S1 Data. Datasets.**
(XLSX)

## Acknowledgments

Special thanks to William Kwara, Nkaiseng Ngwane, and Malefesane Soai for their technical assistance during data collection. I also appreciate the support from Dr Llang Maama Maime, the Program Manager of the National TB Control Programme of Lesotho, the Ethics Committee of the Lesotho Health Services, and all TB patients who participated in this study. Finally, I am grateful to Dr. Heidi, Dr. Raymond Panas, Dr. Chinaro Kennedy, and Brown Carey for their invaluable guidance and thorough manuscript review.

## Author contributions

**Conceptualization:** Jerry Yakubu Yahaya.

**Data curation:** Jerry Yakubu Yahaya.

**Formal analysis:** Jerry Yakubu Yahaya.

**Funding acquisition:** Jerry Yakubu Yahaya.

**Investigation:** Jerry Yakubu Yahaya.

**Methodology:** Jerry Yakubu Yahaya.

**Project administration:** Jerry Yakubu Yahaya.

**Resources:** Jerry Yakubu Yahaya.

**Software:** Jerry Yakubu Yahaya.

**Supervision:** Jerry Yakubu Yahaya.

**Validation:** Jerry Yakubu Yahaya.

**Visualization:** Jerry Yakubu Yahaya.

**Writing – original draft:** Jerry Yakubu Yahaya.

**Writing – review & editing:** Jerry Yakubu Yahaya.

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
