## [Decision Letter · Decision Letter 0]

PGPH-D-24-02643

Sociodemographic Determinants of Multidrug-Resistant Tuberculosis in Lesotho: A Case-Control Study

Dear Dr. Yahaya,

Thank you for submitting your manuscript to PLOS Global Public Health. After careful consideration, we feel that it has merit but does not fully meet PLOS Global Public Health’s publication criteria as it currently stands. Therefore, we invite you to submit a revised version of the manuscript that addresses the points raised during the review process.

We look forward to receiving your revised manuscript.

Kind regards,

Sachin Atre, Ph.D.

Academic Editor

Journal Requirements:

1. We ask that a manuscript source file is provided at Revision. Please upload your manuscript file as a .doc, .docx, .rtf or .tex.

2.  Please provide an Author Summary. This should appear in your manuscript between the Abstract (if applicable) and the Introduction, and should be 150–200 words long. The aim should be to make your findings accessible to a wide audience that includes both scientists and non-scientists. Sample summaries can be found on our website under Submission Guidelines:

https://journals.plos.org/globalpublichealth/s/submission-guidelines#loc-parts-of-a-submission.

Additional Editor Comments (if provided):

1. The general introduction can be cut down and specific details on MDR-TB need to be added. 

2. Objectives and research questions can be merged in one paragraph. Please refer earlier published manuscripts how they are organised.

3. It will be useful to specify categorical variables with reference to age and the gender (male and female) should be shown separately. Odds ratios need to presented separately along with the reference variable. The data has a significant number of women than the men. Can authors explain the reasons for this gender difference? Are there more women than men affected with TB?

4. Discussion: Following sentence needs to be reworded. Instead of writing that the income is a protective factor, authors can write that patients belonged to higher socio-economic class are less likely to get MDR-TB infection.

4. The table on socio-demographic characteristics need to be presented before the regression analysis.

Reviewers' comments:

Reviewer's Responses to Questions

**Comments to the Author**

1. Does this manuscript meet PLOS Global Public Health’s publication criteria?

Reviewer #1: Partly

Reviewer #2: No

2. Has the statistical analysis been performed appropriately and rigorously?

Reviewer #1: Yes

Reviewer #2: Yes

3. Have the authors made all data underlying the findings in their manuscript fully available (please refer to the Data Availability Statement at the start of the manuscript PDF file)?

Reviewer #1: No

Reviewer #2: Yes

4. Is the manuscript presented in an intelligible fashion and written in standard English?

Reviewer #1: Yes

Reviewer #2: Yes

Reviewer #1: Overall Impression : The manuscript investigates the sociodemographic determinants of multidrug-resistant tuberculosis (MDR-TB) in Lesotho. The study employs a case-control design and logistic regression analysis to identify significant factors associated with MDR-TB. The manuscript is well-structured and clearly written.

Strengths :

1.Relevant research question: The study addresses a critical gap in understanding the sociodemographic factors influencing MDR-TB in Lesotho.

2.Robust study design: The case-control design and logistic regression analysis are suitable for identifying associations between sociodemographic factors and MDR-TB.

3.Large sample size : The study includes a substantial sample of 306 participants, which enhances the generalizability of the findings.

Weaknesses :

1.Self-reported data: The reliance on self-reported data may introduce recall bias or social desirability bias.

2.Limited generalizability: The study's findings may not be generalizable to other countries or settings due to Lesotho's unique socioeconomic context.

3.Lack of control for confounding variables: The study does not account for potential confounding variables, such as alcohol use, tobacco use, and treatment adherence.

Methodological Concerns :

1.Sampling strategy: The manuscript does not provide detailed information on the sampling strategy used to select participants.

2.Data collection methods: The study relies on phone surveys and face-to-face interviews, which may introduce differences in data quality.

3.Variable definitions: Some variables, such as caregiver support, are not clearly defined, which may lead to ambiguity in interpretation.

Statistical Concerns :

1.Model specification: The logistic regression model does not account for potential interactions between variables.

2.Model fit: The manuscript does not report on the model's goodness-of-fit statistics.

3.Variable selection: The study does not provide a clear rationale for the selection of variables included in the logistic regression model.

Conclusion:

The manuscript provides valuable insights into the sociodemographic determinants of MDR-TB in Lesotho. However, the study has several methodological and statistical limitations that should be addressed to enhance the validity and generalizability of the findings. With revisions to address these concerns, the manuscript has the potential to contribute significantly to the understanding of MDR-TB in resource-constrained settings.

Recommendations :

1.Clarify sampling strategy and data collection methods.

2.Provide clear definitions for variables.

3.Account for potential confounding variables.

4.Report on model fit statistics and variable selection rationale.

5.Consider addressing potential biases and limitations.

Reviewer #2: There is a growing consensus that advancing tuberculosis control in low- and middle-income countries requires not only investments in TB control programs, diagnostics, and treatment but also targeted efforts to address the social determinants of the disease. This manuscript highlights this critical issue. However, the research methodology requires further justification, and the discussion section would benefit from the inclusion of more recent references to strengthen its arguments.

**Do you want your identity to be public for this peer review?** For information about this choice, including consent withdrawal, please see our Privacy Policy

Reviewer #1: No

Reviewer #2: No

---

## [Decision Letter · Decision Letter 1]

PGPH-D-24-02643R1

Sociodemographic Determinants of Multidrug-Resistant Tuberculosis in Lesotho: A Case-Control Study

Dear Dr. Yahaya,

Thank you for submitting your manuscript to PLOS Global Public Health. After careful consideration, we feel that it has merit but does not fully meet PLOS Global Public Health’s publication criteria as it currently stands. Therefore, we invite you to submit a revised version of the manuscript that addresses the points raised during the review process.

We look forward to receiving your revised manuscript.

Kind regards,

Sachin Atre, Ph.D.

Academic Editor

Journal Requirements:

Additional Editor Comments (if provided):

1. Line 4: "This study aimed to identify key sociodemographic determinants

associated with MDR-TB among adult TB patients in Lesotho".

This sentence in the abstract is repeated.

2. “Ethics statement” should be placed at the beginning of Methods section.

Reviewers' comments:

Reviewer's Responses to Questions

**Comments to the Author**

Reviewer #1: All comments have been addressed

Reviewer #2: All comments have been addressed

publication criteria?

Reviewer #1: Yes

Reviewer #2: Yes

3. Has the statistical analysis been performed appropriately and rigorously?

Reviewer #1: Yes

Reviewer #2: Yes

4. Have the authors made all data underlying the findings in their manuscript fully available (please refer to the Data Availability Statement at the start of the manuscript PDF file)?

Reviewer #1: Yes

Reviewer #2: Yes

5. Is the manuscript presented in an intelligible fashion and written in standard English?

Reviewer #1: Yes

Reviewer #2: Yes

Reviewer #1: Accepted !

Reviewer #2: The author has carefully considered and incorporated the feedback provided in the previous round of review. The revisions are satisfactory and the changes made have improved the clarity and overall quality of the manuscript. I have no further suggestions, and I recommend the revised version for acceptance.

**Do you want your identity to be public for this peer review?** For information about this choice, including consent withdrawal, please see our Privacy Policy

Reviewer #1: **Yes: ** Chaitali Nikam

Reviewer #2: No

---

## [Editor Report · Decision Letter 2]

Sociodemographic Determinants of Multidrug-Resistant Tuberculosis in Lesotho: A Case-Control Study

PGPH-D-24-02643R2

Dear 

We are pleased to inform you that your manuscript 'Sociodemographic Determinants of Multidrug-Resistant Tuberculosis in Lesotho: A Case-Control Study' has been provisionally accepted for publication in PLOS Global Public Health.

Best regards,

Sachin Atre, Ph.D.

Academic Editor